# Calculating Tumor Volume Using Three-Dimensional Models in Preoperative Soft-Tissue Sarcoma Surgical Planning: Does Size Matter?

**DOI:** 10.3390/jcm12237242

**Published:** 2023-11-22

**Authors:** Jasmijn D. Generaal, Haye H. Glas, Jan F. Ubbels, Marc G. Stevenson, Marijn A. Huijing, Barbara L. van Leeuwen, Lukas B. Been

**Affiliations:** 1Department of Surgery, Division of Surgical Oncology, University Medical Center Groningen, University of Groningen, Hanzeplein 1, 9713 GZ Groningen, The Netherlands; b.l.van.leeuwen@umcg.nl (B.L.v.L.); l.b.been@umcg.nl (L.B.B.); 2Department of Maxillofacial Surgery, 3D Lab, University Medical Center Groningen, University of Groningen, Hanzeplein 1, 9713 GZ Groningen, The Netherlands; h.h.glas@umcg.nl; 3Department of Radiation Oncology, University Medical Center Groningen, University of Groningen, Hanzeplein 1, 9713 GZ Groningen, The Netherlands; j.f.ubbels@umcg.nl; 4Department of Surgery, Isala Hospital, Dr. van Heesweg 2, 8025 AB Zwolle, The Netherlands; m.g.stevenson@isala.nl; 5Department of Plastic Surgery, University Medical Center Groningen, University of Groningen, Hanzeplein 1, 9713 GZ Groningen, The Netherlands; m.a.huijing@umcg.nl

**Keywords:** soft-tissue sarcoma, extremity, three-dimensional, three-dimensional virtual surgical planning, flap reconstruction, limb-salvage therapy

## Abstract

This feasibility study aims to explore the use of three-dimensional virtual surgical planning to preoperatively determine the need for reconstructive surgery following resection of an extremity soft-tissue sarcoma. As flap reconstruction is performed more often in advanced disease, we hypothesized that tumor volume would be larger in the group of patients that had undergone flap reconstruction. All patients that were treated by surgical resection for an extremity soft-tissue sarcoma between 1 January 2016 and 1 October 2019 in the University Medical Center Groningen were included retrospectively. Three-dimensional models were created using the diagnostic magnetic resonance scan. Tumor volume was calculated for all patients. Three-dimensional tumor volume was 107.8 (349.1) mL in the group of patients that had undergone primary closure and 29.4 (47.4) mL in the group of patients in which a flap reconstruction was performed, *p* = 0.004. Three-dimensional tumor volume was 76.1 (295.3) mL in the group of patients with a complication following ESTS treatment, versus 57.0 (132.4) mL in patients with an uncomplicated course following ESTS treatment, *p* = 0.311. Patients who had undergone flap reconstruction had smaller tumor volumes compared to those in the group of patients treated by primary closure. Furthermore, a larger tumor volume did not result in complications for patients undergoing ESTS treatment. Therefore, tumor volume does not seem to influence the need for reconstruction. Despite the capability of three-dimensional virtual surgical planning to measure tumor volume, we do not recommend its utilization in the multidisciplinary extremity soft-tissue sarcoma treatment, considering the findings of the study.

## 1. Introduction

Soft-tissue sarcomas (STSs) are rare malignancies originating from mesenchymal tissues and responsible for 700–800 new cases in the Netherlands annually [1,2,3]. There are approximately 80 different histological subtypes of STS, and these subtypes can result in different clinical characteristics. Around 50% of all STSs occur in the extremities, with another 40% occurring in the trunk and head/neck region combined and 10% occurring in the retroperitoneum [4]. The imaging modality of choice for an extremity soft-tissue sarcoma (ESTS) is contrast-enhanced magnetic resonance (MR) imaging, followed by (image guided) core needle biopsy for pathologic diagnosis [1,5,6,7]. Contrast-enhanced MR imaging is the optimal modality to characterize a soft-tissue sarcoma, as it provides high soft-tissue contrast and allows for accurate assessment of lesion size, morphology and relationship to other structures [8,9,10,11,12].

The mainstay of treatment for ESTS is limb-salvage therapy (LST), consisting of surgical resection and radiotherapy (RT) [13,14,15,16]. Patients are preferably treated with preoperative RT instead of postoperative RT for its favorable characteristics, such as smaller radiation fields and a lower total dosage of RT. As a result, limb function has improved in the long-term, although both preoperative and postoperative RT result in morbidity and decreased functional outcome [7,13,14,15,17]. One disadvantage from preoperative RT is that it more frequently results in wound complications, which occur in 35% of patients compared to 17% for postoperative RT. Tumor size has also been identified as one of the predictors of wound complications [18]. Since the 5-year survival for localized ESTS patients has increased to 70–75% [19,20], finding options to reduce treatment-induced morbidity in the ESTS population has become more relevant.

Reconstructive surgery, such as pedicled or free flaps, are used in LST to allow optimal oncological resection as well as coverage of vital structures, limb function preservation and optimization of aesthetic outcome [21,22,23,24]. Reconstructive surgery is considered when a lack of tissue is expected to achieve primary closure and/or enable adequate coverage of vital structures. In selected cases, it is indicated to provide functional recovery through nerve/vessel reconstruction or replacement of muscle groups. The reconstruction of tissue defects is an option to reduce treatment-related morbidity in the ESTS population [16,23,25,26,27,28]. Flap reconstruction is more frequently performed in distal extremities, older patients and more advanced disease [28]. Now, the indication for reconstructive surgery is based on the expert opinion of the oncologic and plastic surgeon and is decided per individual patient, taking into account multiple patient- and treatment-related factors. Regularly, the necessity of reconstructive surgery can only be determined during the operation. Anticipating the need for reconstructive surgery may lead to specific positioning and draping of the patient to optimize the surgical site for both tumor removal and subsequent reconstruction. Furthermore, it impacts the overall duration of the procedure and affects the allocation of resources, such as the availability of specific instruments and expertise of the surgical team. The current study is aimed at improving preoperative patient selection to ensure that patients receive comprehensive and well-planned treatment that addresses both ESTS removal and subsequent reconstructive needs.

In this retrospective cohort study, the feasibility of three-dimensional (3D) virtual surgical planning to preoperatively determine the need for reconstructive surgery following resection of an ESTS was explored. Three-dimensional virtual surgical planning enables quantifying tumor- and patient-specific factors, such as tumor volume. Tumor volume was hypothesized to be a significant factor influencing the indication for reconstructive surgery as it is performed more often in more advanced disease [28]. Furthermore, a larger tumor size has been identified as a predictor of wound complications [18], and flap reconstruction is performed to prevent morbidity in a population at high risk for developing wound complications [21].

## 2. Materials and Methods

All patients ≥ 18 years old that were treated by surgical resection (with wide margins of 2 cm healthy tissue) for a localized or locally advanced STS in the upper or lower extremity from 1 January 2016 until 1 October 2019 in the University Medical Center Groningen (UMCG) were included in this study retrospectively. The UMCG is one of six sarcoma specialist centers in the Netherlands and the only academic center in northern Netherlands, which has a population of three million people. The Institutional Review Board of the UMCG approved this retrospective study (case number 2019.504). Informed consent was waived. Some patients had undergone two resections in the aforementioned period of time: a primary resection for an ESTS and a resection for a recurrence. In these cases, patients were included for the separate resections. Patients with atypical lipomatous tumors were excluded, as treatment strategy is different for these tumors and the need for reconstructive surgery is seldom. Patients were excluded when diagnostic magnetic resonance (MR) images were absent or tumors extended past the cranial limit of the acetabulum or head of the humerus.

All patients included in the study underwent an identical diagnostic work-up at a specialized tertiary center with a dedicated musculoskeletal tumor board that includes a radiologist. All MR examinations were performed by using a 1.5 T scanner (Avanto; Siemens; Erlangen, Germany). Tumors were planned in Mimics (Materialise Mimics Medical version 20.0; Leuven, Belgium) using the contrast-enhanced (Gadolinium) T1-weighted sequence in the axial plane of the diagnostic MR scan in all patients. For ESTS patients, this is the standard protocol for tumor assessment. Three-dimensional (3D) virtual surgical planning of the tumor was performed by segmentation, based on signal intensity of different tissues in the MR scans. For segmentation, a slice thickness of 4 mm was used. The primary step in the process was to plan the tumor volume. First, a mask was set up by selecting high signal intensity tumor tissue. Subsequently, this mask was split to divide tumor tissue from arteries/veins that were also dyed with the contrast agent. For fine tuning of the mask, it was smoothed, holes were filled and it was in some cases opened, closed or eroded by a few voxels. When the mask contained all tumor tissue, a 3D model was calculated. The 3D model was adjusted by using a smooth factor of 0.4, and the model was wrapped 1 mm in all directions. Three-dimensional tumor volume was extracted from the properties of the model. See Figure 1 for 3D tumor volume.

Data on tumor and patient characteristics were extracted from the patient files, such as type of closure. Tumor localization and depth were determined by assessing the diagnostic MR scan. Tumor depth was noted as superficial or deep, according to the Tumor Nodus Metastasis (TNM) classification system. Tumors were superficial when located exclusively above the fascia without invasion of the fascia. Deep tumors were located exclusively beneath the fascia or superficial to the fascia with invasion of or through the fascia. Wound complications were scored according to the Clavien-Dindo classification [29]. Grade I complications were scored when there was any deviation from the normal course, without the need for pharmacological or surgical intervention. Bedside openings of infections were scored as a Clavien I. Grade II complications required pharmacological treatment, for instance antibiotics. When patients had undergone hyperbaric oxygen therapy, this was scored as a Clavien II. Grade III complications needed surgical or radiological intervention. Grade IIIa was scored when the intervention did not take place under general anesthesia, and Grade IIIb was marked when the intervention was performed under general anesthesia. Grade IV referred to life-threatening complications, whereas grade V referred to complications resulting in the death of a patient.

Statistical analyses were performed using SPSS^®^ Version 28.0 (IBM SPSS Statistics for Windows, Version 23.0 Arnouk, NY, USA: IBM Corp). Numerical variables were compared by using the Kruskal–Wallis test. *p*-values of ≤0.05 indicated statistical significance.

## 3. Results

### 3.1. Population

Ninety-five patients were initially included in the study. Twenty-five patients were excluded for the following reasons. Fifteen patients were excluded for having an atypical lipomatous tumor, and the other ten exclusions were MR related. In five patients, the tumor was not pictured completely on the MR images. Due to insufficient quality MR images, another two patients were excluded. In one patient, the tumor extended past the extremity into the pelvis. Diagnostic MR images were not available for one patient. One patient was excluded for having two tumors that were pictured in separate axial MR images. See Figure 2.

### 3.2. Patient, Tumor and Treatment Characteristics

Fifty patients (71.4%) had undergone primary closure following ESTS resection. In 13 (18.6%) patients, a skin graft was used and in 7 (10%) patients a flap reconstruction was performed. The median age of patients was 68.9 (IQR = 28) years, 66.6 (IQR = 15) years and 71.6 (IQR = 27) years in the primary closure group, skin graft group and flap reconstruction group, respectively. In the primary closure group, ESTS were located in the upper arm in 5 (10.0%) patients, the lower arm in 3 (6.0%) patients, the upper leg in 36 (72.0%) patients and the lower leg in 6 (12.0%) patients. ESTS were located in the upper arm in 2 (15.4%) patients, the lower arm in 2 (15.4%) patients, the upper leg in 4 (30.8%) patients and the lower leg in 5 (38.5%) patients in which a skin graft was used to close a defect after resection. One (14.3%) patient had undergone a flap reconstruction for an ESTS in the upper arm. Flap reconstruction was performed for ESTS located in the lower arm in 2 (28.6%) patients, the upper leg in 2 (28.6%) patients and the lower leg in 2 (28.6%) patients. ESTS were deep in 41 (82.0%) patients in which defects were primarily closed, compared to 4 (30.8%) deep ESTS in the skin graft group and 4 (57.1%) deep ESTS in the flap reconstruction group. In the group of patients that underwent flap reconstruction, in two (28.6%) patients a skin excision was required to obtain safe margins. ESTS were high grade in 28 (56.0%) patients in the primary closure group. There were 11 (84.6%) high grade ESTS in the skin graft group and 4 (57.1%) in the flap reconstruction group. Preoperative RT was part of the treatment strategy in most patients: 34 (68.0%) patients in the primary closure group, 8 (61.5%) patients in the skin graft group and 5 (71.4%) patients in the flap reconstruction group. Another six (12.0%) patients in the primary closure group and one (7.7%) patient in the skin graft group had undergone postoperative RT. None of the patients in which a flap reconstruction was performed were treated by postoperative RT. Treatment consisted of hyperthermic isolated limb perfusion, preoperative RT, and surgery in four (8.0%) patients in the primary closure group and in three (42.9%) patients in the flap reconstruction group. In the primary closure group, in 34 (68.0%) patients treatment was uncomplicated. Eleven (22.0%) patients had a Grade II complication, three (6.0%) patients had a Grade IIIa complication and two (4.0%) patients had a Grade IIIb complication. Treatment was complicated for four (37.1%) patients in the skin graft group; in these patients the complications were scored as a Grade I, II, IIIa and III. Two (28.6%) patients had a complication in the flap reconstruction group, these were scored as Grade I and Grade IIIb. See Table 1 for patient, tumor and treatment characteristics.

### 3.3. Three-Dimensional Virtual Surgical Planning: Tumor Volume

Three-dimensional tumor volume was 107.8 (IQR = 349.1) mL in the group of patients that had undergone primary closure, 22.0 (IQR = 48.4) mL in the group of patients with skin grafts and 29.4 (IQR = 47.4) mL in the group of patients in which a flap reconstruction was performed, *p* = 0.004. Three-dimensional tumor volume was 76.1 (IQR = 295.3) mL in the group of patients with a complication following ESTS treatment, versus 57.0 (IQR = 132.4) mL in patients with an uncomplicated course following ESTS treatment, *p* = 0.311.

## 4. Discussion

In this study, it was explored whether preoperative assessment of the indication for reconstructive surgery could be improved by using 3D virtual surgical planning. The aim was to improve preoperative patient selection for flap reconstruction and thereby reduce treatment-related morbidity following ESTS treatment. Tumor volume was hypothesized to be a significant factor influencing flap reconstruction as it is more often performed in more advanced disease. Also, a larger tumor size has been identified as a predictor of wound complications [18], and flap reconstruction is usually performed to prevent morbidity in a population at high risk for developing wound complications [21]. When comparing 3D tumor volume between groups of closure types, 3D tumor volume was smallest in the group of patients that had undergone flap reconstruction (29.4 mL vs. 107.4 mL in the primary closure group). Furthermore, 3D tumor volume did not significantly differ between patients with a complication versus without a complication (76.1 mL vs. 57.0 mL) following STS treatment. It is concluded that tumor volume does not seem to influence the indication for flap reconstruction. The need for reconstructive surgery following ESTS resection is presumably multifactorial. Available literature reports that flap reconstruction is more often performed in elderly patients and patients with advanced disease (requiring radiotherapy or chemotherapy), and tumors are more likely to be located in the distal extremities [21].

Complications following surgical resection are multifactorial adverse events, due to dead space, tension on wound edges and impaired vascularity. The use of flaps counteracts all these results of surgical resection, as it fills up dead space, recovers vascularity in preoperative irradiated tissue and releases tension on wound edges [26]. However, as flap reconstruction is characterized by being complex (vascular) surgery, due to prolonged operation time, more blood loss and impaired perfusion, reconstructive surgery can also lead to postoperative morbidity. Research concerning this topic has been contradictory, but recent publications argue that reconstructive surgery can be superior to primary closure in selected patients [21,23,25,26,27]. In this study, the group of patients that had undergone flap reconstruction for a tissue defect was smaller than expected; in only seven patients (10%) flap reconstructions were performed. In other studies, reported proportions of patients with flap reconstructions range from 15 to 30% [21,22,30]. In our center, the indication for reconstructive surgery remains a subject of debate. The alternative to flap reconstruction is closing the tissue defect using a skin graft and thereby avoiding the anastomotic issues inherent to flap reconstructions. Vascularization of the wound bed is an important determinant for survival, yet skin grafts can survive on relatively avascular sites [31,32]. Some plastic surgeons had a more active attitude towards flap reconstruction than others; this may provide an explanation for the difference in the use of flap reconstruction in our center versus reported literature.

Reported complication rates for ESTS patients that have undergone resection combined with flap reconstruction are 38–50% [21,33]. Although patients at high risk of developing a wound complication were represented in the flap reconstruction group, the complication rate of patients that underwent flap reconstruction versus primary closure did not differ significantly in the study, being 28.6% vs. 32.0%, respectively. This confirms the idea that the transfer of vascularized flaps after ESTS resection may prevent complications in selected patients.

This study is limited by being retrospective and by the limited number of patients. However, almost four years of patients that were surgically treated for an ESTS were included, and the population size was adequate to investigate the implementation of a new modality. Validation of the 3D virtual surgical planning could not be performed, as we relied on the use of contrast-enhanced MR images, which are recognized as the gold standard in ESTS imaging. Furthermore, as the feasibility of a new modality was determined in the current study, validation falls beyond its scope. MR imaging has demonstrated its effectiveness for identifying the location, grade and anatomical relationships of ESTS [8,9,10,11,34]. Surgeons also rely on these diagnostic MR images to assess operability of the tumor and to strategize the surgical approach [10]. These findings provide support for using the diagnostic MR images as a basis for creating preoperative 3D models. The preoperatively planned 3D tumor volume could not be compared to the volume of the resected tumor as the study was retrospective in nature and tumor volume is not measured routinely ex vivo. Three-dimensional models were verified for accuracy by a trained physician.

To our knowledge, this study is the first to explore the use of 3D virtual surgical planning in ESTS treatment. The use of 3D virtual surgical planning has been trending in trauma surgery, maxillofacial surgery and neurosurgery as more optimal preoperative planning has been shown to lead to an improved outcome, such as shorter operation time [35]. Three-dimensional virtual surgical planning is not without costs, and the clinical benefits should be weighed before implementation into everyday clinical practice. This study does not add to the evidence promoting the use of 3D virtual surgical planning in patients with ESTS as tumor volume does not seem to influence the indication for flap reconstruction. Future research should be focused on determining the patient- and treatment-related predictors of flap reconstruction to improve preoperative patient selection and the multidisciplinary extremity-soft tissue sarcoma management.

## Figures and Tables

**Figure 1 jcm-12-07242-f001:**
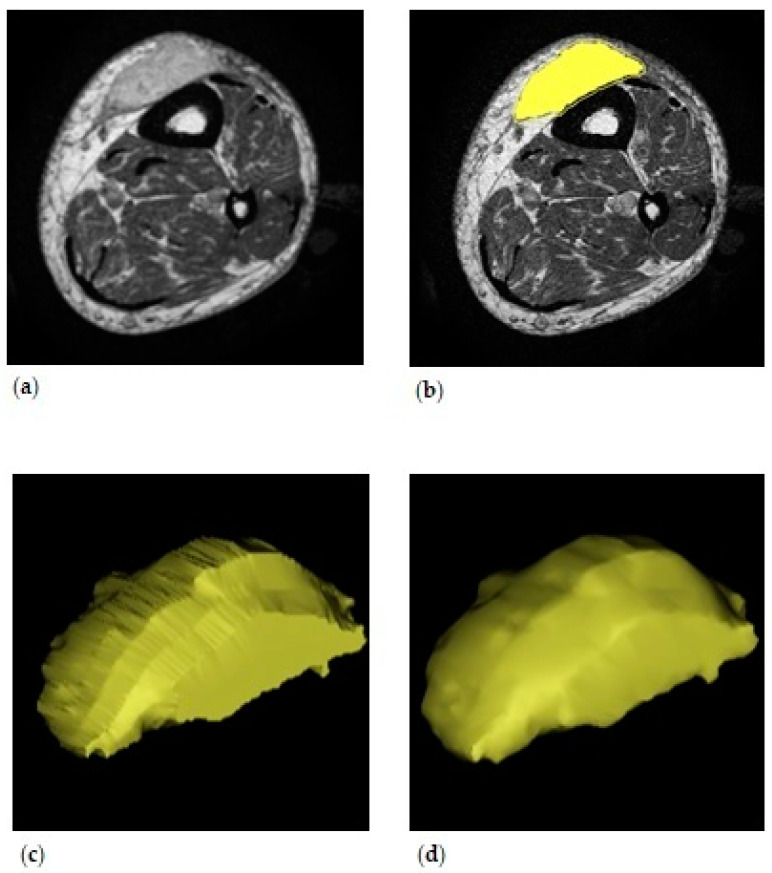
Three-dimensional volume of the left lower leg. (**a**) The original contrast-enhanced T1-weighted MR scan in the axial plane. (**b**) Planned tumor volume. The line around the yellow mask indicates the wrapped area. (**c**) The three-dimensional model generated from the planned tumor volume in the axial plane. (**d**) The final three-dimensional model.

**Figure 2 jcm-12-07242-f002:**
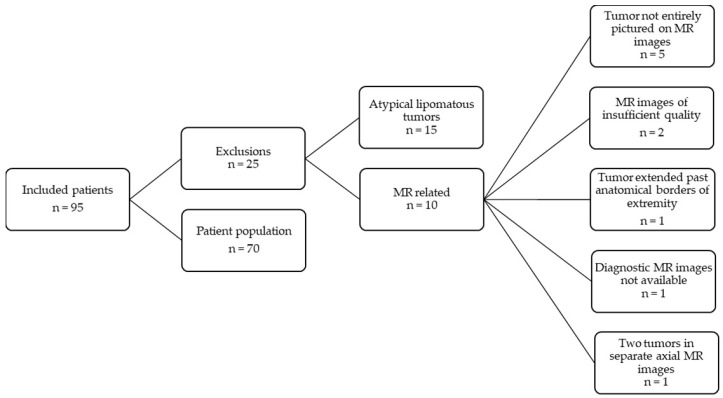
Flowchart of the patient population.

**Table 1 jcm-12-07242-t001:** Patient, tumor and treatment characteristics (*n* = 70).

		Primary Closure *n* = 50	Skin Graft *n* = 13	Flap Reconstruction *n* = 7
Age (years)		68.9 (28)	66.6 (15)	71.6 (27)
BMI ^1^ (kg/m^2^)		26.4 (4.9)	25.1 (5.3)	28.5 (9.6)
Tumor localization	Upper arm	5 (10.0)	2 (15.4)	1 (14.3)
	Lower arm	3 (6.0)	2 (15.4)	2 (28.6)
	Upper leg	36 (72.0)	4 (30.8)	2 (28.6)
	Lower leg	6 (12.0)	5 (38.5)	2 (28.6)
Tumor depth	Superficial	9 (18.0)	9 (69.2)	3 (42.9)
	Deep	41 (82.0)	4 (30.8)	4 (57.1)
Tumor grade	Borderline	1 (2.0)	1 (7.7)	0 (0)
	Low	6 (12.0)	0 (0)	1 (14.3)
	High	28 (56.0)	11 (84.6)	4 (57.1)
Radiotherapy	None	10 (20.0)	4 (30.8)	2 (28.6)
	Preoperative RT ^2^	34 (68.0)	8 (61.5)	5 (71.4)
	Postoperative RT	6 (12.0)	1 (7.7)	0 (0)
PRS ^3^	No	46 (92.0)	13 (100)	4 (57.1)
	Yes	4 (8.0)	0 (0)	3 (42.9)
Wound complications	None	34 (68.0)	9 (62.9)	5 (71.4)
	Grade I	0 (0)	1 (7.7)	0 (0)
	Grade II	11 (22.0)	1 (7.7)	1 (14.3)
	Grade IIIa	3 (6.0)	1 (7.7)	0 (0)
	Grade IIIb	2 (4.0)	1 (7.7)	1 (14.3)
	Grade IV and V	0 (0)	0 (0)	0 (0)

Data are presented as *n* (%) or median (IQR). Abbreviations. ^1^ BMI: Body Mass Index. ^2^ RT: radiotherapy. ^3^ PRS: Hyperthermic isolated limb perfusion, preoperative radiotherapy and surgery.

## Data Availability

The datasets generated during and/or analyzed during the current study are available from the corresponding author on reasonable request. The data are not publicly available due to ethical restrictions. Data were pseudo-anonymized by using an identification code, but place of treatment and several dates related to the patient could make it identifiable.

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
