# Peer review of "Calculating Tumor Volume Using Three-Dimensional Models in Preoperative Soft-Tissue Sarcoma Surgical Planning: Does Size Matter?"

_jcm, 2023, doi:10.3390/jcm12237242_

Round 1

Reviewer 1 Report

Comments and Suggestions for Authors

Abstract

- The conclusion is not entirely clear. Please rewrite and set a clear target.

Introduction

- Line 42: MRI is a highly valuable imaging modality to assess STS's size or volume. On the one hand, please extend the paragraph a little bit. On the other hand, your references regarding MRI are quite old. Please change them to the following:

1.      https://doi.org/10.1007/s00238-020-01669-1

2.      https://doi.org/10.1016/j.suronc.2020.08.023

3.      https://doi.org/10.1186/s12885-021-08113-y

4.      https://doi.org/10.1177/02841851211008381

5.      https://doi.org/10.2478/raon-2021-0007

6.      https://doi.org/10.5114/pjr.2020.94687

Methods

- What MRI systems were used?

- Were there any errors in MRI postprocessing?

- How can you perform quantitative analysis using SPSS? I guess you mean statistical analysis?

Results

- Flowchart: Please include why the cases were excluded.

- Please edit your statistics and add some p values where necessary.

Discussion

- Please add a paragraph about MRI and please add the reference mentioned under “Introduction” where fitting.

Comments on the Quality of English Language

The English needs minor revision and proof-read.

Reviewer 2 Report

Comments and Suggestions for Authors

This is a feasibility study that aims to explore efficacy of three-dimensional virtual surgical planning to anticipate the necessity of reconstruction surgery after excision of an extremity soft-tissue sarcoma.

Although this study is the first to explore the use of 3D virtual surgical planning in ESTS treatment, some modifications are required to be published.

#1. Please spell out ESTS.

#2. In lines 67-68, introduction, it is stated that Regularly, the necessity for reconstructive surgery can only be determined during the operation which can lead to logistic issues and possibly undertreatment of patients that may benefit from flap reconstruction.

I believe the authors anticipate situations in which reconstructive surgery may be necessary and prepare prior to surgery, even if the chances of reconstruction are low. Does the preoperative anticipation of reconstructive surgery affect patient positioning, draping, or total surgery time?

#3. In this study, although the authors hypothesized that tumor volume was a significant factor to influence flap reconstruction, 3D tumor volume turned out to be much smaller in primary closure group than in skin graft or flap reconstruction group. In lines 229-232, discussion, it is stated that The need for reconstructive surgery following ESTS resection is presumably multifactorial. Available literature reports that flap reconstruction is more often performed in elderly patients, advanced disease (requiring radiotherapy or chemotherapy) and tumors are more likely to be located in the distal extremities. 

I think need for reconstructive surgery can depend on the surgical procedure. Did the authors perform wide excision of ESTS with a safety margin in all cases? If so, to what extent? Were cases in which skin excision was performed to guarantee a safety margin included? If a tumor is located deeper than the fascia, classified as tumor depth of deep, I would think a skin excision would not be necessary even if the tumor is large, which would contribute to primary closure.

Comments on the Quality of English Language

Please spell out ESTS.

Round 2

Reviewer 1 Report

Comments and Suggestions for Authors

Your revision tremendously improved the quality of your manuscript. Thanks!